# Outborn newborns drive birth asphyxia mortality rates—An 8 year analysis at a rural level two nursery in Uganda

Anna Hedstrom[1‡]*, James Nyonyintono[2‡], Paul Mubiri[3], Hilda Namakula Mirembe[4], Brooke Magnusson[4], Josephine Nakakande[2], Molly MacGuffie[4], Mushin Nsubuga[2], Peter Waiswa[3], Harriet Nambuya[5], Maneesh Batra[1]

1 Departments of Pediatrics and Global Health, University of Washington, Seattle, Washington, United States of America, 2 Kiwoko Hospital, Kiwoko, Uganda, 3 Makerere University School of Public Health, Kampala, Uganda, 4 Adara Development, Edmonds, Washington, United States of America, 5 Department of Pediatrics, Jinja Regional Referral Hospital, Jinja, Uganda

‡ AH and JN are co-first authors on this work.
* hedstrom@uw.edu

**Data Availability Statement:** The dataset is included as supplementary material.

## Abstract

Birth asphyxia is a leading cause of global neonatal mortality. Most cases occur in low- and middle- income countries and contribute to half of neonatal deaths in Uganda. Improved understanding of the risk factors associated with mortality among these patients is needed. We performed a retrospective cohort study of a clinical database and report maternal demographics, clinical characteristics and outcomes from neonates with birth asphyxia at a Ugandan level two unit from 2014 through 2021. "Inborn" patients were born at the hospital studied and "outborn" were born at another facility or home and then admitted to the hospital studied. Doctors assigned the patient's primary diagnosis at death or discharge. We performed a Poisson model regression of factors associated with mortality among patients with asphyxia. The study included 1,565 patients with birth asphyxia and the proportion who were outborn rose from 26% to 71% over eight years. Mortality in asphyxiated patients increased over the same period from 9% to 27%. Factors independently associated with increased death included outborn birth location (ARR 2.1, p<0.001), admission in the year 2020 (ARR 2.4, p<0.05) and admission respiratory rate below 30bpm (RR 3.9, p<0.001), oxygen saturation <90% (ARR 2.0, p<0.001) and blood sugar >8.3 mmol/L (RR 1.7, p<0.05). Conversely, a respiratory rate >60bpm was protective against death (ARR 0.6, p<0.05). Increased birth asphyxia mortality at this referral unit was associated with increasing admission of outborn patients. Patients born at another facility and transferred face unique challenges. Increased capacity building at lower-level birth facilities could include improved staffing, training and equipment for labor monitoring and newborn resuscitation as well as training on the timely identification of newborns with birth asphyxia and resources for transfer. These changes may reduce incidence of birth asphyxia, improve outcomes among birth asphyxia patients and help meet global targets for newborn mortality.

**Funding:** The author(s) received no specific funding for this work.

**Competing interests:** The authors have declared that no competing interests exist.

## Introduction

Currently, 2.4 million newborns die each year- leaving the world far from reaching the Sustainable Development Goal reduction to 12 neonatal deaths or less per 1,000 live births by the year 2030 [1, 2]. The vast majority of all neonatal deaths occur in resource limited settings (RLS) including 43% in Sub-Saharan Africa [2]. Up to 29% of newborn mortality is due to birth asphyxia- making it the second leading cause of newborn death globally [3–5]. Birth asphyxia occurs when a baby does not establish breathing at birth and is therefore at risk for brain damage due to a period of inadequate oxygen supply [6]. It is the leading cause of cerebral palsy and also responsible for up to 63% of cognitive impairment in those without cerebral palsy [7]. Birth asphyxia causes a loss of 53 million disability adjusted life years per year and this burden is not distributed evenly around the world; rather, it is inversely correlated with the development index of each country [3].

Most of the deaths and lifelong disabilities due to birth asphyxia can be averted through interventions and programs which prioritize quality facility-based childbirth and care of sick newborns [8–10]. Although the standard of care for patients with asphyxia in high resource settings includes therapeutic hypothermia which can reduce mortality and morbidity, this therapy has not been associated with similar benefits in RLS [11, 12]. This difference in response to therapeutic hypothermia may be due to differing co-morbidities among populations in RLS which may affect response to treatment [13]. Beyond provision of timely essential newborn care and resuscitation, along with supportive care, evidence-based therapies for the treatment of birth asphyxia in RLS are lacking. Therefore evidence around birth asphyxia mortality is needed to guide facility development in RLS in the areas of obstetric care, newborn resuscitation, timely transfer and specialized newborn care for the asphyxiated neonate [14, 15].

The incidence of birth asphyxia and survival rates vary dramatically globally and minimal data from RLS are available to guide improvement of treatments [2–4, 16, 17]. Modeling exercises to estimate the burden of birth asphyxia have traditionally been derived from tertiary urban referral facilities and extrapolated to lower-level facilities. This has resulted in widely varying estimates of incidence, prevalence, and risk factors. For example, while one pre-COVID pandemic model reported the age-adjusted incidence of asphyxia to be stable in the lowest resourced areas between 1990 and 2019 [16], another model found the age-standardized prevalence rate increased by 150% during the same period [3]. Identification of factors associated with survival of babies with birth asphyxia from the range of facility levels within a health system are necessary for identifying opportunities for improving outcomes [1, 4, 18, 19].

Kiwoko Hospital (KH) in central Uganda serves a large rural catchment area and has a well-developed level two newborn care unit. The hospital's neonatal prospective database includes key demographic and clinical metrics on all admissions since 2014 making it a unique longitudinal repository of neonates who received facility-based care in a level 2 nursery. Our study aimed to find risk factors for mortality among newborns admitted with birth asphyxia.

## Methods

### Study design

Retrospective cohort study of a clinical database.

### Study population

Analysis in this study was restricted to neonates (28 days of age or less on admission) admitted to the special care nursery for whom the birth location was recorded. There were no other exclusion criteria.

## COVID period

No admission criteria changed during the COVID pandemic. The early COVID period (April through September 2020) saw a 14% decrease in admissions but an increased proportion of admission with birth asphyxia (22% vs. 15%) [20].

## Setting

Kiwoko Hospital (KH) is a rural, private, not-for-profit general hospital that acts as a secondary level care newborn referral center for three districts (total population 1,000,000) in north central Uganda. Between 2011–2016, the neonatal mortality rate in the region was 30 per 1,000 live births and 75% of births took place in facilities [21]

This well-developed level two newborn care unit opened in 2001 and is a regional leader in the care of small and sick newborns [22]. It has capacity to care for 38 babies at a time, and has more than 1,100 admissions annually. The unit has operational funding from outside organizations including Adara Development and families pay for a limited portion of the care their baby receives, as able. There are 36 nurses on staff in the newborn unit with a median nurse to patient ratio of 1:9, as well as one assigned medical officer and one pediatric physician. Electricity is constantly available with the help of a stand-by generator. As of 2019/2020, there were 2562 deliveries, 38% via cesarean section [23]. Stillbirth rate at the hospital has remained constant during the period of study- median 37.5 per 1,000 births (IQR 36.9, 40.1) according to hospital records. Admissions via ambulance to KH are assisted by government primary health care subsidies. Outborn patients are treated in the same unit and receive the same care as those born at KH.

## Clinical care of patients with birth asphyxia

Throughout the period of study, care provided to patients with birth asphyxia included thermoregulation via overhead warmer, incubators and kangaroo mother care, intravenous hydration including use of IV pumps, nasogastric, cup and breast feeding, phototherapy, antibiotics, anticonvulsants, neurodevelopmental care including promoting flexed positioning, blood transfusion, basic laboratory services including complete blood count and electrolytes, CPAP therapy [24, 25] andintermittent as well as continuous pulse oximetry as needed. The unit does not have capacity for surfactant, mechanical ventilation, parenteral nutrition, central lines, blood culture, coagulation measurement, electro encephalogram, MRI or therapeutic hypothermia, and these therapies are not routinely available even via transfer to the national referral hospital located two hours travel by vehicle. Close monitoring of patients included care by nurses with an average of 9 patients to 1 nurse. Continuous physician coverage was available as needed and rounds were lead daily by a pediatrician or medical officer.

## Data collection and analysis

The hospital has maintained a digital database of all nursery admissions as part of routine health systems data since late 2012. Patient and maternal information are entered on a designated bedside form by nurses on admission. After discharge, the data entry team extracts additional information from the medical file including treatments and final diagnosis designation by unit physician at death, discharge, or transfer. Data were manually entered into Epi Info version 7 [26] through 2020, and more recently into RedCap [27]. The study team was given a deidentified version of this dataset on 3 May, 2022.

## Outcome

Primary diagnosis was assigned at death, discharge, or transfer by the doctor on duty. Survival to discharge from the unit excluded patients who were referred, left the hospital against medical advice and those with missing discharge status as their ultimate survival is unknown.

## Study variables

The diagnosis of birth asphyxia was at the discretion of the discharging physician, and utilized available information including 5-minute Apgar score <7, poor tone and/or presence of clinically suspected seizure. The unit did not utilize a standardized exam of patients with risk of asphyxia such a Sarnat staging or Thompson scoring.

Although in other settings low apgars have been associated with poor outcomes [28, 29], reported apgars in this dataset are unreliable, and therefore high apgar scores are not routinely used as exclusion criteria for asphyxia by discharging physicians. Of note, in settings with advanced diagnostics newborns suffering from intrapartum-related events can be diagnosed with neonatal encephalopathy or hypoxic ischemic encephalopathy to better classify their pathology [30]. However, given these diagnostic capacities are lacking in many RLS, this paper will use the term asphyxia.

Patients were classified as 'inborn' if they were born at KH. 'Outborn' refers to patients born at another facility, home or on the way to the hospital. When birthweight was unknown, admission weight was used if patient was admitted within 3 days of birth (n = 1,369). Gestational age is not reliably recorded throughout the period of study and so is not reported.

We investigated the missing data variable by variable. The proportion of missing data ranged from 1% to 30%. No method of imputing missing data was used since missing data was not considered to be missing at random. Variables were left out of multivariate analyses if reported on less than 80% of patients (n = 1,252) or if they represented a transport/treatment decision (i.e. transfer via ambulance, use of phototherapy, CPAP or blood transfusion).

## Statistical methods

All statistical analyses were performed using STATA v.17 (Stata Corporation, College Station, Texas, USA). Descriptive analysis included frequencies (proportions) for categorical variables and medians, and interquartile range for continuous variables. Continuous variables including respiratory rate, temperature, oxygen saturation among others were grouped into clinically relevant categories [31]. We assessed the association between the covariates and inborn/outborn status and between covariates and discharge status using a Chi-Square test statistic for categorical variable and Mann Whitney test for continuous variables.

We estimated diagnosis specific mortality rate over time and corresponding 95% CIs using a Poisson regression model accounting for exposure time from admission to discharge/death. We stratified our analysis by inborn/outborn status to produce stratum-specific mortality estimates and corresponding 95% CIs for only newborns with birth asphyxia at death or discharge. The mortality rates were plotted to show trends over time and to compare between the inborn and outborn newborns with birth asphyxia.

We fitted a Poisson regression model with robust variance and accounting for the exposure time from admission to discharge/death to examine the factors predictive of birth asphyxia associated mortality. Unadjusted and adjusted risk ratios (RR) were reported. All potential factors were tested for collinearity using variance inflation factors (VIF). When factors were found to be collinear or correlated, one was dropped from the analysis. We compared two models using Akaike Information Criterion (AIC) and Bayesian Information Criterion (BIC) to examine two models and selected the most parsimonious.

We further stratified our analysis by inborn/outborn status to examine if the factors predictive of birth asphyxia associated mortality were different for those born at KH (inborn) and those newborns born elsewhere and referred to KH (outborn). Outborn patients were previously found to have higher mortality than inborn patients [32].

This research has adhered to the *STROBE guidelines* for cross sectional studies [33].

### Ethical considerations

Human Subjects Approval was obtained from Makerere University School of Public Health Institutional Review Board (protocol number 917) and approved by the Uganda National Council for Science and Technology (registration number SS813ES). The data were fully anonymized before accessed by the research team and the ethics committee waived the requirement for informed consent. The University of Washington institutional review board designated this as an exempt study.

## Results

### All admissions

Nine thousand and nine patients were admitted between 2014 and 2021 (Fig 1). We excluded 146 patients who were >28 days of age on admission and 86 who had an unrecorded birth location. Forty seven percent of admissions were outborn. Primary diagnoses at death or discharge included prematurity (4,035, 46%), infection (2,184, 25%), birth asphyxia (1,565, 18%) and other (993, 11%).

### Mortality among major diagnoses

Overall unit mortality was 12% and does not include 3% who were referred for care elsewhere. Mortality rate per 1,000 admissions to the unit due to each diagnosis were: birth asphyxia [209

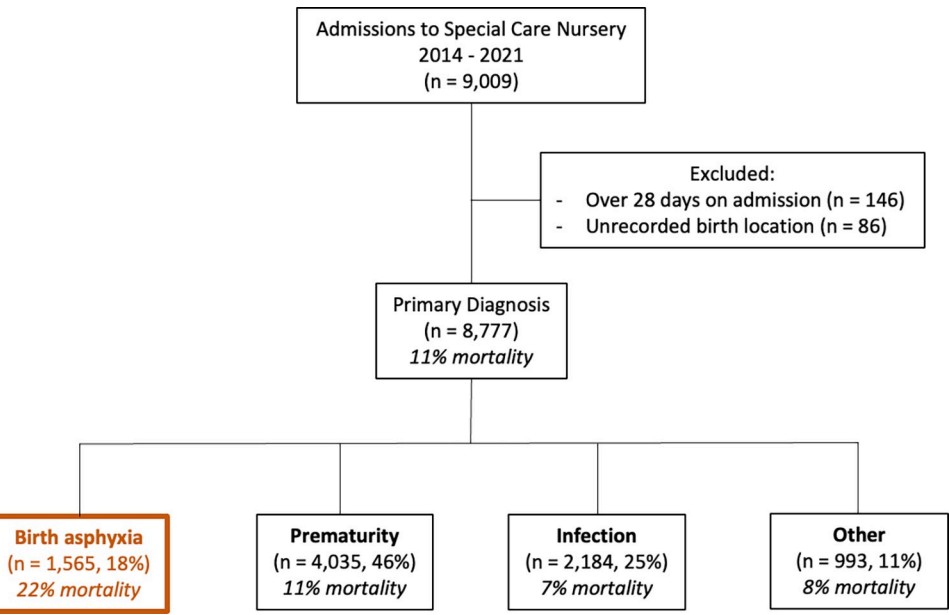

**Fig 1. Neonatal patient population admitted from 2014–2021.** Our primary analysis sample includes the 1,565 patients with a diagnosis of asphyxia. Mortality rates among asphyxia patients (22%) was higher than all other diagnoses.

per 1,000 (95% CI 183–236)], prematurity [96 per 1,000 (95% CI 86–107)] and infection [68 per 1,000 (95% CI 57–80)]. The trend in these mortality rates over time are shown in S1 Fig. Overall rates of mortality were higher in asphyxiated patients than the other two diagnoses. Peak mortality for each diagnosis was in 2020 during the early COVID-19 pandemic. While mortality rates for prematurity and infection have trended back towards baseline since 2020, asphyxia mortality rates instead rose steadily over the period of study from 9% to 27%.

## Birth asphyxia patients

Our primary analysis sample includes the 1,565 patients with a diagnosis of asphyxia. The proportion of admissions with asphyxia remained stable from 2014 through 2021, around 18%. Fifty one percent (803) were inborn at KH and the remaining were outborn. Outborn admissions with asphyxia were predominantly born at another facility (96.6%, 736) and a small number born at home (3%) or on the way to hospital (0.4%).

The proportion of patients with birth asphyxia who were outborn increased steadily from 26% (42/161) in 2014 to 71% (166/233) in 2021. (S2 Fig) This trend was not observed for diagnoses of prematurity or infection.

Outborn birth asphyxia admissions had a higher mortality rate than those inborn [273 per 1,000 live births (95% CI: 224–322) vs. 132 per 1,000 live births (95% CI: 99–166), *p< 0.001*)]. Fig 2 shows the trend in the differential mortality over time between these groups- reaching statistical significant during 2019–2021.

## Outborn vs. inborn characteristics

As shown in Table 1, mothers of outborn birth asphyxia patients were more likely to reside outside Kiwoko's district (67% vs. 59%, *p* = 0.001). Outborn babies were more likely to arrive at Kiwoko via ambulance than were mothers of patients later admitted for asphyxia after birth

**Fig 2. Mortality among outborn birth asphyxia patients.** Remained higher than those who were inborn, and this difference was statistically significant between 2019–2021. The decreased mortality in 2017 is unexplained and likely due to a difference in how data was recorded in that period.

**Table 1.** Characteristics for neonates with birth asphyxia by birth location.

| | All patients (N = 1,565) n (%) | Inborn (N = 803) n (%) | Outborn (N = 762) n (%) | |
|---|---|---|---|---|
| **Maternal Demographics** | | | | |
| **Mother's age (years)** (n = 1374) | | | | |
| <20 | 330 (24.0) | 145 (22.0) | 185 (25.9) | |
| 20–35 | 942 (68.6) | 460 (69.8) | 482 (67.4) | |
| >35 | 102 (7.4) | 54 (8.2) | 48 (6.7) | |
| **Gravida** (n = 1474) | | | | |
| 1 (primigravid) | 612 (45.5) | 292 (39.1) | 320 (44.0) | |
| 2–3 | 494 (33.5) | 256 (34.3) | 238 (32.7) | |
| ≥ 4 | 368 (25.0) | 199 (26.6) | 169 (23.3) | |
| **Antenatal care visits** (n = 1302) | | | | |
| 0 | 3 (0.2) | 3 (0.4) | 0 (0.0) | |
| 1–3 | 564 (43.3) | 289 (45.2) | 275 (41.5) | |
| ≥ 4 | 735 (56.5) | 348 (54.4) | 387 (58.5) | |
| **Maternal district** (n = 1565) | | | | * |
| Within KH district | 578 (36.9) | 327 (40.7) | 251 (32.9) | |
| Outside KH district | 987 (63.1) | 476 (59.3) | 511 (67.1) | |
| **Mode of transport** (n = 1565) | | (maternal) | (newborn) | ** |
| Ambulance | 334 (21.3) | 78 (9.7) | 256 (33.6) | |
| Another vehicle | 1096 (70.0) | 607 (75.6) | 489 (64.2) | |
| Bicycle or foot | 135 (8.6) | 118 (14.7) | 17 (2.2) | |
| **At Birth** | | | | |
| **Type of birth attendant** (n = 1565) | | | | ** |
| Doctor | 511 (32.6) | 379 (47.2) | 132 (17.3) | |
| Midwife/nurse | 1035 (66.1) | 422 (52.6) | 613 (80.5) | |
| TBA, family member or other | 19 (1.2) | 2 (0.2) | 17 (2.2) | |
| **Mode of delivery** (n = 1565) | | | | ** |
| Vaginal | 1083 (69.2) | 441 (54.9) | 642 (84.2) | |
| Caesarean section | 482 (30.8) | 362 (45.1) | 120 (15.8) | |
| **Meconium-stained fluid** (n = 1039) | | | | ** |
| No | 585 (56.3) | 333 (51.5) | 252 (64.1) | |
| Yes | 454 (43.7) | 313 (48.5) | 141 (35.9) | |
| **Cried at birth** (n = 1388) | | | | |
| No | 1146 (82.6) | 581 (72.3) | 565 (83.2) | |
| Yes | 242 (17.4) | 128 (15.9) | 114 (16.8) | |
| **Apgar—median [IQR]** | | | | |
| 1 minute (n = 1227) | 5 (4–6) | 5 (4–6) | 6 (5–7) | |
| 5 minute (n = 1202) | 6 (5–8) | 6 (5–7) | 7 (5–8) | |
| **Infant characteristics** | | | | |
| **Sex** (n = 1565) | | | | |
| Female | 608 (38.8) | 314 (39.1) | 294 (38.6) | |
| Male | 957 (61.2) | 489 (60.9) | 468 (61.4) | |
| **Multiple birth** (n = 1560) | | | | * |
| Singleton | 1488 (95.4) | 754 (94.1) | 734 (96.7) | |
| Multiple birth | 72 (4.6) | 47 (5.9) | 25 (3.3) | |
| **Age at admission** (days) (n = 1565) | | | | ** |
| Day of birth | 1410 (90.1) | 793 (98.7) | 617 (81.0) | |

*(Continued)*

**Table 1.** (Continued)

| | All patients (N = 1,565) n (%) | Inborn (N = 803) n (%) | Outborn (N = 762) n (%) | |
|---|---|---|---|---|
| 1–2 days of age | 141 (9.0) | 6 (0.8) | 135 (17.7) | |
| 3+ | 14 (0.9) | 4 (0.5) | 10 (1.3) | |
| **Birthweight** (n = 1546) | | | | |
| < 2.5kg | 162 (10.5) | 91 (11.4) | 71 (9.5) | |
| 2.5–4.49kg | 1369 (88.5) | 703 (87.9) | 666 (89.3) | |
| 4.5kg+ | 15 (1.0) | 6 (0.8) | 9 (1.2) | |
| **Admission Vital Signs** | | | | |
| **Temperature** (°C) (n = 1537) | | | | ** |
| < 36.5 | 898 (58.4) | 604 (**76.7**) | 294 (39.2) | |
| 36.5–37.9 | 480 (31.2) | 179 (22.7) | 301 (40.1) | |
| ≥ 38 | 159 (10.3) | 4 (0.5) | 155 (**20.7**) | |
| **Respiratory rate** (bpm) (n = 1379) | | | | |
| < 30 | 42 (3.1) | 22 (3.1) | 20 (3.0) | |
| 30–60 | 638 (46.3) | 350 (49.4) | 288 (43.0) | |
| > 60 | 699 (50.7) | 337 (47.5) | 362 (54.0) | |
| **Oxygen saturation** (%) (n = 1417) | | | | |
| < 90 | 556 (36.2) | 298 (41.5) | 258 (36.9) | |
| ≥ 90 | 861 (60.8) | 420 (58.5) | 441 (63.1) | |
| **Blood sugar** (mmol/L) (n = 1441) | | | | * |
| < 2.6 | 96 (6.7) | 39 (5.2) | 57 (8.2) | |
| 2.6–8.3 | 1058 (73.4) | 546 (73.0) | 512 (73.9) | |
| >8.3 | 287 (19.9) | 163 (21.8) | 124 (17.9) | |
| **Clinical Course** | | | | |
| **Therapies received:** | | | | |
| Phototherapy (n = 1545) | 283 (18.3) | 130 (16.3) | 153 (20.4) | * |
| Blood transfusion (n = 1555) | 22 (1.4) | 11 (1.4) | 11 (1.4) | |
| Bubble CPAP (n = 1543) | 277 (18.0) | 110 (13.9) | 167 (22.2) | ** |
| **Died during admission** | 339 (21.7) | 103 (12.8) | 236 (31.0) | ** |

*p <0.05

** <0.001

at Kiwoko (34% vs. 9.7%, $p<0.001$). When compared with inborns, outborn birth asphyxia patients were less likely to be delivered by a doctor (17% vs. 47%, $p<0.001$) or via cesarean section (16% vs. 45%, $p<0.001$). Outborns were also less likely to have report of meconium-stained amniotic fluid (36% vs. 48%, $p<0.001$), although this metric was reported in less than 80% of patients and will be not included in multivariate analysis. Median birthweight was 3030g (IQR 2780, 3400) for outborn patients vs. 3100g (IQR 2770, 3400) for inborn patients, $p = 0.587$.

Outborn birth asphyxia patients were more likely admitted after the day of birth (19% vs. 1.3%, $p<0.001$), and to have a high temperature (20.7% vs. 0.5%, $p<0.001$) and a low blood sugar (8% vs. 5%, $p = 0.022$) when compared to inborns. Inborn patients, in contrast, were more likely to have a low temperature (77% vs. 39%, $p<0.001$). Once admitted, outborn patients were more likely to be treated with phototherapy (20% vs. 16%, $p = 0.046$), and bubble CPAP (22% vs. 14%, $p<0.001$) than inborns. Outborn birth asphyxia patients had a mortality of 31% vs. 13% mortality for inborns, $p< 0.001$.

### Patient characteristics by mortality outcome

We compared birth asphyxia patients who died during admission vs. those that survived to discharge (S1 Table). Babies that died were less likely to have been delivered via cesarean section (26% vs. 32%, $p = 0.02$), but more likely outborn (70% vs. 43%, $p<0.001$) and <2.5kg at birth (15% vs. 9%, $p<0.005$) compared with those that survived. These patients who died (or their laboring moms) were more likely to have arrived via ambulance (34% vs. 18%, $p<0.001$). Median birthweight of surviving birth asphyxia patients was higher than those who died [3,100grams (IQR 2,800, 3,400) vs 3,000grams (IQR 2,700, 3,380), $p< 0.001$].

Vital signs on admission among those who died were more likely to include high temperature (17% vs. 8%, $p<0.001$), low respiratory rate <30bpm (9% vs. 2%, $p<0.001$), low oxygen saturation (59% vs. 34%, $p<0.001$) and high blood sugar (31% vs. 17%, $p<0.001$). Neonates with birth asphyxia who died during admission were less likely to be treated with phototherapy (10% vs. 21%, $p< 0.001$) but more likely to be treated with bubble CPAP (45% vs. 10%, $p< 0.001$) compared to those who survived.

### Risk factors for mortality

Table 2 shows the Poisson regression model for risk factors independently associated with mortality among birth asphyxia patients. Risk factors included outborn birth location [ARR 2.1 (95% CI 1.5–3.1)], respiratory rate below 30bpm [ARR 3.9 (95% CI 1.9–7.8), oxygen saturation <90% [ARR 2.0 (95% CI 1.4–2.7)], blood sugar >8.3 mmol/L [ARR 1.7 (95% CI 1.1–2.5)] and admission in 2020 [ARR 2.4 (95% CI 1.1–5.1)]. Respiratory rate greater than 60bpm was associated with decreased death (ARR 0.5, $p<0.05$). S2 Table shows risk factors for mortality separated by inborn and outborn patients.

## Discussion

We report steadily increasing mortality among birth asphyxia patients in a rural Ugandan level two nursery from 9% to 27% over eight years. This is associated with increasing proportion of outborn asphyxia patients over time whose risk of mortality is double that of inborn patients. The poor outcomes among outborn neonates with asphyxia have been reported in several settings including this unit during the 2005–2008 period [32, 34–41] One referral facility in Tanzania reported delivery at another facility among birth asphyxia patients was associated with 3 times higher risk of neurological impairment [28].

Other factors independently associated with mortality included low temperature which as previously been associated with more severe asphyxia and mortality [39, 40, 42, 43]. More severely asphyxiated babies likely underwent prolonged resuscitation and may have had inadequate thermal support during resuscitation and transport as has been described in many RLS [39, 40]. Conversely, high temperature was reported in 20% of outborn patients. Although data was not available to assess, this could be a marker of concurrent infection in these patients. Low respiratory rate [28] and oxygen saturation [28, 40, 44] both indicate respiratory failure. High respiratory rate, however, was protective against death and suggests more mild asphyxia or another disease pathway. High blood sugar was associated with mortality and suggests physiologic stress and possible infection [40]. Lastly, birth in 2020 corresponds to other reports of increased neonatal mortality due to travel restrictions during the early COVID-19 period in Uganda [20, 45–47].

Increased admissions of outborn birth asphyxia patients over time could be driven by more asphyxia cases in the region and/or increased referrals of patients. As our study does not include a population-based sample, we are unable to discern between the two. Although age-adjusted global incidence of asphyxia has decreased over time, overall number of deaths have

**Table 2. Independent risk factors for death among birth asphyxia patients.**

| | | Crude RR (95% CI) | Adjusted RR (95% CI) |
|---|---|---|---|
| Birth location | Inborn | ref | ref |
| | Outborn | 1.7 (1.4-2.2) ** | **2.1 (1.5-3.1) ** ** |
| Mode of delivery | Vaginal | ref | |
| | Caesarean | 0.9 (0.7-1.1) | |
| Sex | Male | ref | |
| | Female | 1.03 (0.8-1.3) | |
| Birth weight (grams) | <2500 | 1.3 (0.9-1.8) | 1.4 (0.8-2.3) |
| | ≥2500 | ref | ref |
| Day of age on admission | Day of birth | ref | |
| | 1-2 days | 1.01 (0.8-1.3) | |
| | 3+ days | 0.7 (0.4-1.3) | |
| Temperature (˚C) on admission | <36.5 | 1.4 (1.1-1.9) * | ^ |
| | 36.5-37.9 | ref | |
| | 38+ | 1.8 (1.3-2.7) ** | |
| Respiratory rate (bpm) on admission | <30 | 3.5 (1.9-6.3) ** | **3.9 (1.9-7.8) ** ** |
| | 30-60 | ref | ref |
| | >60 | 0.7 (0.5-0.9) ** | 0.6 (0.4-0.8) * |
| Oxygen saturation (%) on admission | < 90 | 2.4 (1.8-3.1) ** | **2.0 (1.4-2.7) ** ** |
| | ≥ 90 | ref | ref |
| Blood sugar (mmol/L) on admission | <2.6 | 1.5 (0.97-2.4) | 1.4 (0.8-2.6) |
| | 2.6-8.3 | ref | ref |
| | >8.3 | 2.1 (1.6-2.8) ** | **1.7 (1.1-2.5) ** * |
| Year of admission | 2014 | ref | ref |
| | 2015 | 1.4 (0.8-2.5) | 1.8 (0.8-4.3) |
| | 2016 | 1.6 (0.9-2.8) | 1.1 (0.5-2.9) |
| | 2017 | 1.1 (0.6-2.0) | 1.1 (0.5-2.6) |
| | 2018 | 1.9 (1.1-3.2) * | 2.0 (0.9-4.5) |
| | 2019 | 1.9 (1.1-3.2) * | 2.0 (0.9-4.4) |
| | 2020 | 1.9 (1.2-3.2) * | **2.4 (1.1-5.1) ** * |
| | 2021 | 1.8 (1.1-3.0) * | 1.8 (0.8-3.9) |

Ref = reference group.

*p<0.05

**p<0.001.

^Given collinearity between temperature and outborn status, temperature was not included in multivariate (Poisson) regression.

increased steadily since 1990, suggesting there were more babies to care for with asphyxia [16]. Komakech [48] reported that Ugandan birth asphyxia rates rose between 2017 and early 2020, however this included the early COVID-19 pandemic period travel restrictions, which were particularly impactful on safe delivery and newborn care [20, 49]. The trend towards increased facility birth and training in Helping Babies Breathe may have led to more effective resuscitation and initial survival of asphyxiated newborns [50–52]. Accordingly, there may be an increasing trend to identify and refer asphyxia patients to a higher level of care due to a regional emphasis on importance of neonatal care and resuscitation.

Increased mortality may be because outborn patients with birth asphyxia have presented sicker over time. This may be due to limited lower-level and private facility readiness to handle more complicated deliveries and support asphyxiated newborns [15, 51]. Additionally,

improved newborn resuscitation at these facilities may have increased initial survival of severely asphyxiated newborns to transport. Other factors contributing to sick transports may be delayed care including identification of high-risk pregnancy symptoms, timely labor management and diagnosis of the asphyxiated newborn [53, 54]. Although increased funding started in 2019 for ambulance transport through the Ministry of Health Results Based Financing scheme, difficulty in neonatal transfer from lower-level facilities was exacerbated recently by the COVID-19 pandemic [20, 51]. Transports without thermoregulatory, glucose or breathing support could have impacted the condition of the patient upon admission to KH.

Limitations of this study include the diagnosis of birth asphyxia by the physician was done without availability of continuous peripartum monitoring or resuscitation data, standardized scoring for encephalopathy, blood gas measurement, EEG monitoring and magnetic resonance imaging. The dataset was subject to missing variables and antenatal data was limited to reporting by parents. We did not check for interactions between variables. Lastly, the data are from a single-center with above-average resources for the region and therefore we are unable to generalize these trends to other sites in or outside of Uganda. Strengths of this study include registry of each admission over eight years, creating a large repository of patients which was well powered to detect trends and risk factors. The study period includes periods pre COVID and after the initial COVID period.

## Conclusions

Given the global focus on regionalization of neonatal care and referral of the sick newborn to improve care, it is important to understand the survival disadvantage among transferred neonates with birth asphyxia [55, 56]. Scaling up quality, impactful care to prevent and treat birth asphyxia in lower-level facilities requires increased awareness of unique features of this disease and delegation of improved resources.

As described by Elizabeth Ayebare [51], clinicians could be assisted by the generation of national guidelines specifying urgent pre-referral treatments for mothers with complicated labor and asphyxiated newborns [10, 51, 57, 58]. These should identify which patients require referral and highlight the importance of initial supportive care including dextrose containing fluids, temperature management, phenobarbital, antibiotics as indicated and respiratory support [13].

Building capacity and quality at lower-level facilities should use quality improvement methodology and provision of a comprehensive service package to include [4, 48, 51, 59–63]:

1. Identification of champions at each facility to train others and create a critical competent mass to care for newborns [64]

2. Maintaining appropriate nursing and doctor staffing ratios and avoidance of staff rotation away from the unit [65]

3. Ongoing mentorship by experienced clinicians [60, 64]

4. Improving intrapartum monitoring and resuscitation practice [30, 54]

5. Provision of equipment for and training on neonatal resuscitation [15, 52, 66]

6. Ensuring timely and safe transport via ambulance including monitoring/thermal support. (ex TOPS score [39])

7. Standardization of feedback to referring facility after receipt of a sick newborn [53]

Additionally, all facilities could benefit from a strengthened data collection system and standardization of diagnosis and cause of death by providers to ensure accuracy [2, 4, 53]. Given

the risk of cerebral palsy and cognitive impairment among asphyxiated newborns, higher level facilities need to work to ensure these babies have appropriate neurodevelopmental follow-up and support [7, 15, 57, 67, 68].

Decreasing the incidence of asphyxia and improving survival of these patients can't be pinpointed to any one intervention, but overall advancement in peripartum care with focus on adequate staffing and training in neonatal resuscitation would likely improve outcomes.

## Supporting information

**S1 Fig. Mortality rates among major diagnoses over time.** (A- birth asphyxia, B- prematurity and C- infection). Peak mortality for each diagnosis was in 2020 during the early COVID-19 pandemic. While mortality has returned to baseline for premature infants (B) and is decreasing back towards baseline for neonates with infection (C), the mortality rates for birth asphyxia patients (A) has steadily increased over time.
(TIF)

**S2 Fig. Trends in birth location by major diagnoses over time.** The proportion of birth asphyxia patients who were outborn increased over the period of study from 26% to 71%. This trend towards increasing admission of outborn patients was not seen in other major diagnoses.
(TIF)

**S1 Table. Characteristics for neonates with birth asphyxia by survival to discharge.**
(DOCX)

**S2 Table. Risk factors for death among birth asphyxia patients by birth location.**
(DOCX)

**S1 Data.**
(XLS)

## Acknowledgments

Dr. Becca Jones.

## Author Contributions

**Conceptualization:** Anna Hedstrom, James Nyonyintono, Maneesh Batra.

**Data curation:** Anna Hedstrom, Hilda Namakula Mirembe, Brooke Magnusson, Josephine Nakakande.

**Formal analysis:** Anna Hedstrom, Paul Mubiri.

**Methodology:** Anna Hedstrom, Paul Mubiri, Hilda Namakula Mirembe, Brooke Magnusson, Mushin Nsubuga, Peter Waiswa, Harriet Nambuya.

**Project administration:** Anna Hedstrom, Maneesh Batra.

**Resources:** Molly MacGuffie.

**Supervision:** Anna Hedstrom, James Nyonyintono, Hilda Namakula Mirembe, Brooke Magnusson, Josephine Nakakande.

**Validation:** Molly MacGuffie, Mushin Nsubuga, Harriet Nambuya.

**Writing – original draft:** Anna Hedstrom.

**Writing – review & editing:** James Nyonyintono, Paul Mubiri, Hilda Namakula Mirembe, Brooke Magnusson, Molly MacGuffie, Mushin Nsubuga, Peter Waiswa, Harriet Nambuya, Maneesh Batra.

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
