## [Decision Letter · Decision Letter 0]

30 Aug 2023

PGPH-D-23-01252

Outborn Newborns Drive Birth Asphyxia Mortality Rates- an 8 Year Analysis at a Rural Level Two Nursery in Uganda

Dear Dr. Hedstrom,

Thank you for submitting your manuscript to PLOS Global Public Health. After careful consideration, we feel that it has merit but does not fully meet PLOS Global Public Health’s publication criteria as it currently stands. Therefore, we invite you to submit a revised version of the manuscript that addresses the points raised during the review process.

We look forward to receiving your revised manuscript.

Kind regards,

Miquel Vall-llosera Camps

Staff Editor

Journal Requirements:

Reviewers' comments:

Reviewer's Responses to Questions

**Comments to the Author**

1. Does this manuscript meet PLOS Global Public Health’s publication criteria? Is the manuscript technically sound, and do the data support the conclusions? The manuscript must describe methodologically and ethically rigorous research with conclusions that are appropriately drawn based on the data presented.

Reviewer #1: Yes

Reviewer #2: Yes

2. Has the statistical analysis been performed appropriately and rigorously?

Reviewer #1: Yes

Reviewer #2: Yes

3. Have the authors made all data underlying the findings in their manuscript fully available (please refer to the Data Availability Statement at the start of the manuscript PDF file)?

Reviewer #1: Yes

Reviewer #2: Yes

4. Is the manuscript presented in an intelligible fashion and written in standard English?

Reviewer #1: Yes

Reviewer #2: Yes

5. Review Comments to the Author

Reviewer #1: I would recommend adding some data about the standard criteria for the diagnosis of severe birth asphyxia (pH, BE, Apgar score, etc.).

Also, give more details about the COVID-19 pandemic period and what was your admission guideline during that period.

You can also be more specific about the guideline used in these patients and what would be the means of improving the outcome.

Reviewer #2: Perinatal asphyxia is responsible for a large number of NICU admissions and neonatal deaths in low- and middle-resource countries. His work is in line with other publications on the African continent.

Improvements in prenatal and perinatal care are needed to reduce the mortality of asphyxiated newborns.

This study helps to know the local reality and makes it possible to compare it with other similar African studies to adopt measures that allow expanding the information and taking action within the global health approach.

Cavallin F, Menga A, Brasili L, Maziku D, Azzimonti G, Putoto G, Trevisanuto D. Factors associated with mortality among asphyxiated newborns in a low-resource setting. J Matern Fetal Neonatal Med. 2022 Mar;35(6):1178-1183. doi: 10.1080/14767058.2020.1743670. Epub 2020 Mar 25. PMID: 32212882.

6. PLOS authors have the option to publish the peer review history of their article (what does this mean?). If published, this will include your full peer review and any attached files.

**Do you want your identity to be public for this peer review?** For information about this choice, including consent withdrawal, please see our Privacy Policy.

Reviewer #1: No

Reviewer #2: No

---

## [Decision Letter · Decision Letter 1]

12 Oct 2023

Outborn Newborns Drive Birth Asphyxia Mortality Rates- an 8 Year Analysis at a Rural Level Two Nursery in Uganda

PGPH-D-23-01252R1

Dear Dr. Hedstrom,

We are pleased to inform you that your manuscript 'Outborn Newborns Drive Birth Asphyxia Mortality Rates- an 8 Year Analysis at a Rural Level Two Nursery in Uganda' has been provisionally accepted for publication in PLOS Global Public Health.

Best regards,

Julia Robinson

Executive Editor

Reviewer Comments (if any, and for reference):

Reviewer's Responses to Questions

**Comments to the Author**

1. If the authors have adequately addressed your comments raised in a previous round of review and you feel that this manuscript is now acceptable for publication, you may indicate that here to bypass the “Comments to the Author” section, enter your conflict of interest statement in the “Confidential to Editor” section, and submit your "Accept" recommendation.

Reviewer #1: All comments have been addressed

Reviewer #2: All comments have been addressed

2. Does this manuscript meet PLOS Global Public Health’s publication criteria? Is the manuscript technically sound, and do the data support the conclusions? The manuscript must describe methodologically and ethically rigorous research with conclusions that are appropriately drawn based on the data presented.

Reviewer #1: Partly

Reviewer #2: Yes

3. Has the statistical analysis been performed appropriately and rigorously?

Reviewer #1: Yes

Reviewer #2: Yes

4. Have the authors made all data underlying the findings in their manuscript fully available (please refer to the Data Availability Statement at the start of the manuscript PDF file)?

Reviewer #1: Yes

Reviewer #2: Yes

5. Is the manuscript presented in an intelligible fashion and written in standard English?

Reviewer #1: Yes

Reviewer #2: Yes

6. Review Comments to the Author

Reviewer #1: I don't believe there is enough improvement compared to the first version of the manuscript.

The changes should be highlighted to be more easily identified during the peer-reviewing process.

I recommend that the authors provide us a version of the manuscript that emphasize the changes.

Reviewer #2: none

7. PLOS authors have the option to publish the peer review history of their article (what does this mean?). If published, this will include your full peer review and any attached files.

**Do you want your identity to be public for this peer review?** For information about this choice, including consent withdrawal, please see our Privacy Policy.

Reviewer #1: No

Reviewer #2: No
